

# A novel histozoic myxosporean, *Enteromyxum caesio* n. sp., infecting the redbelly yellowtail fusilier, *Caesio cuning*, with the creation of the Enteromyxidae n. fam., to formally accommodate this commercially important genus

Mark A. Freeman[1], Tetsuya Yanagida[2] and Àrni Kristmundsson[3]

[1] Department of Biomedical Sciences, Ross University School of Veterinary Medicine, Basseterre, Saint Kitts and Nevis
[2] Laboratory of Veterinary Parasitology, Joint Faculty of Veterinary Medicine, Yamaguchi University, Yoshida, Yamaguchi, Japan
[3] Institute for Experimental Pathology at Keldur, University of Iceland, Reykjavík, Iceland

## ABSTRACT

Gastrointestinal myxosporean parasites from the genus *Enteromyxum* are known to cause severe disease, resulting in high mortalities in numerous species of cultured marine fishes globally. Originally described as *Myxidium* spp., they were transferred to a new genus, *Enteromyxum*, to emphasize their novel characteristics. Their retention in the family Myxidiidae at the time was warranted, but more comprehensive phylogenetic analyses have since demonstrated the need for a new family for these parasites. We discovered a novel *Enteromyxum* in wild fish from Malaysia and herein describe the fourth species in the genus and erect a new family, the Enteromyxidae n. fam., to accommodate them. *Enteromyxum caesio* n. sp. is described infecting the tissues of the stomach in the redbelly yellowtail fusilier, *Caesio cuning*, from Malaysia. The new species is distinct from all others in the genus, as the myxospores although morphologically similar, are significantly smaller in size. Furthermore, small subunit ribosomal DNA sequence data reveal that *E. caesio* is <84% similar to others in the genus, but collectively they form a robust and discrete clade, the Enteromyxidae n. fam., which is placed as a sister taxon to other histozoic marine myxosporeans. In addition, we describe, using transmission electron microscopy, the epicellular stages of *Enteromyxum fugu* and show a scanning electron micrograph of a mature myxospore of *E. caesio* detailing the otherwise indistinct sutural line, features of the polar capsules and spore valve ridges. The Enteromyxidae n. fam. is a commercially important group of parasites infecting the gastrointestinal tract of marine fishes and the histozoic species can cause the disease enteromyxosis in intensive finfish aquaculture facilities. Epicellular and sloughed histozoic stages are responsible for fish-to-fish transmission in net pen aquaculture systems but actinospores from an annelid host are thought to be necessary for transmission to fish in the wild.

Corresponding author
Mark A. Freeman,
mafreeman@rossvet.edu.kn,
mf33bitw@gmail.com

## INTRODUCTION

Myxosporeans from the genus *Enteromyxum* (*Palenzuela, Redondo & Alvarez-Pellitero, 2002*), are all parasites of the gastrointestinal (GI) tract of marine fishes. Currently there are only three species described. The type species, *Enteromyxum scophthalmi*, was described from farmed turbot, *Scophthalmus maximus*, in the Mediterranean (*Branson, Riaza & Alvarez-Pellitero, 1999*; *Palenzuela, Redondo & Alvarez-Pellitero, 2002*), *Enteromyxum leei* was originally described from cultured sea bream *Sparus aurata* in Cyprus and Israel (*Diamant, Lom & Dyková, 1994*), but is now known to infect multiple hosts (*Padrós et al., 2001*; *Picard-Sánchez et al., 2020*), and *Enteromyxum fugu* was described infecting tiger puffer, *Takifugu rubripes*, in Japan (*Tun et al., 2000*). Both *E. leei* and *E. fugu* were originally placed in the genus *Myxidium*, due to similarities in spore morphology to some *Myxidium* spp. In the original description of *Myxidium leei* (syn. *E. leei*), *Diamant, Lom & Dyková (1994)* noted that the spore morphology did not convincingly match that of either *Myxidium* or *Zschokkella*, and his placement in *Myxidium* was therefore tentative. Since then, *Palenzuela, Redondo & Alvarez-Pellitero (2002)* and *Yanagida et al. (2004)* have used ribosomal DNA (rDNA) sequence data to demonstrate that *M. leei* and *Myxidium fugu* group together in a robust clade with *E. scophthalmi* and transferred both species to the genus *Enteromyxum*. It is now considered important to support myxosporean species descriptions with rDNA sequence data, as it has been widely demonstrated that some myxospore morphologies cannot be used to reliably place new myxosporean taxa correctly, with some spore morphotypes sharing features with other genetically distant clades within the Myxosporea (*Freeman & Kristmundsson, 2015*). *Enteromyxum leei* and *E. scophthalmi* are serious pathogens in commercial fish farms, causing enteritis and emaciation disease, or enteromyxosis, resulting in high mortalities (*Ogawa & Yokoyama, 2001*; *Palenzuela, Redondo & Alvarez-Pellitero, 2002*; *Sitjà-Bobadilla & Palenzuela, 2012*). Such disease outbreaks have led to the development of sensitive, non-invasive, molecular screening tools for use on farmed fish in order to help mitigate parasite numbers (*Yanagida et al., 2005*; *Alonso et al., 2015*). Reacting quickly to disease outbreaks and controlling parasite numbers is important with enteromyxosis, as fish to fish transmission is known to occur rapidly in fish farms (*Diamant, 1997*; *Redondo et al., 2002*; *Yasuda et al., 2002*; *Picard-Sánchez et al., 2020*), a phenomenon that is unique to *Enteromyxum* amongst the myxosporeans.

The redbelly yellowtail fusilier, *Caesio cuning* (Bloch, 1791), is a tropical reef-associated fish native to the Indian and Western Pacific Oceans. It inhabits coastal areas, usually over rocky substrates and coral reefs, forming schools in mid-water feeding on zooplankton (*Carpenter, 1987*). They are a commercial species throughout most of their range, caught in traps and nets by local fishers.

In the present study, we describe a fourth species that belongs to the genus *Enteromyxum*, infecting the stomach of the redbelly yellowtail fusilier and erect a new family, the Enteromyxidae n. fam., within the suborder Variisporina Lom et Noble, 1984 to accommodate this distinctive myxosporean genus and complete its dissociation from the Myxidiidae sensu stricto (s.s).

## MATERIALS & METHODS

Five *Caesio cuning* were purchased from the fish market in Pulau Ketam near Port Klang, Selangor, three further *C. cuning* were bought from fishermen from Kilim mangroves [6°46′55.36″N; 99°24′49.16″E], Langkawi, Malaysia. All fish were already dead when purchased and were examined as soon as possible for myxosporean parasites. The GI tract was examined in sections, including the anterior stomach/esophagus, stomach and intestine (anterior, middle and posterior/rectum), by scraping a dissected, saline-washed, tissue sample with a scalpel blade and viewing the preparation under a compound microscope. Images of fresh myxospores were taken using an Olympus CX21 microscope fitted with a Dino-Eye eyepiece camera (Dino-lite, UK). The dimensions of 30 myxospores were calculated using the imaging software package ImageJ (NIH, USA). Infected tissue samples were fixed in 10% buffered formalin for histological examination and GI scrapings, containing myxospores, fixed in 2.5% glutaraldehyde for scanning electron microscope (SEM) as previously described (*Kristmundsson & Freeman, 2013*). In addition, intestinal tissues from farmed Japanese tiger puffer, known to have a single intestinal myxosporean infection for the epicellular species *Enteromyxum fugu*, were prepared for transmission electron microscope (TEM) as previously described (*Freeman & Kristmundsson, 2013*) and viewed on a Jeol JEM-1010 transmission electron microscope at Tokyo University.

Intestinal tissues scrapings containing myxospores were stored in lysis buffer from a GeneMATRIX DNA isolation kit (EURx Poland). Total DNA was extracted using the same kit following the tissue protocol. Small subunit ribosomal DNA (SSU rDNA) was amplified using the general myxosporean primers and methodology described by *Freeman, Yokoyama & Ogawa (2008)* with the addition of the new primer Ent-1100fwd 5′-AGAGTACTCACGCAAGTGAT-3′, designed to be specific to this species of *Enteromyxum* only, and to be used with the reverse primer 18gM (*Freeman, Yokoyama & Ogawa, 2008*).

Positive PCR products were purified using a GeneMATRIX PCR products extraction kit (EURx Poland). Bidirectional DNA sequencing was carried out on PCR products of the correct sizes, using the same primers, and nucleotide BLAST searches undertaken to confirm a parasitic identity. The consensus sequence was obtained by eye using CLUSTAL X and BioEdit (*Thompson et al., 1997*; *Hall, 1999*). CLUSTAL X was used for the initial SSU rDNA sequence alignments of the novel sequence and 28 other myxosporeans. Percentage divergence matrices were constructed from selected aligned taxa in CLUSTAL X using the neighbour-joining method based on the Kimura 2-parameter model (*Saitou & Nei, 1987*). Phylogenetic analyses were performed using the maximum likelihood methodology in PhyML (*Guindon et al., 2010*) with the automatic smart model selection (selection criterion: Akaike Information Criterion (AIC)), running the general time-reversible

substitution model (GTR +G6 +I) with 1000 bootstrap repeats. Bayesian inference (BI) analysis were done using MrBayes v. 3.2 (*Ronquist & Huelsenbeck, 2003*). For the BI analysis, models of nucleotide substitution were generated using MrModeltest v. 2.2 (*Nylander et al., 2004*). The most suitable evolutionary model based on the Akaike information criterion was the general time-reversible model (GTR+I+G). Therefore, settings used for the analysis were as previously noted by *Kristmundsson & Freeman (2013)*, with *Chloromyxum riorajum* (FJ624481) marked as the outgroup. Posterior probability distributions were created using the Markov Chain Monte Carlo (MCMC) method running four simultaneous chains for one million generations.

The electronic version of this article in Portable Document Format (PDF) will represent a published work according to the International Commission on Zoological Nomenclature (ICZN), and hence the new names contained in the electronic version are effectively published under that Code from the electronic edition alone. This published work and the nomenclatural acts it contains have been registered in ZooBank, the online registration system for the ICZN. The ZooBank LSIDs (Life Science Identifiers) can be resolved and the associated information viewed through any standard web browser by appending the LSID to the prefix http://zoobank.org/. The LSID for this publication is: lsid:zoobank.org:pub:A8A08CD0-7323-4F4D-A8A9-E51FF86323CE, with the additional LSID identifiers for acts of nomenclature within the publication: lsid:zoobank.org:act:676E54CE-BF4C-4CEE-BEF1-83EA20E88510 (Enteromyxidae n. fam.); lsid:zoobank.org:act:AF828F1F-E6B9-4477-90FD-DE5C27CDE746 (*Enteromyxum - Palenzuela, Redondo & Alvarez-Pellitero, 2002*); lsid:zoobank.org:act:2DE3D545-79D5-4D16-9932-2FC32E09E69E (*Enteromyxum caesio* n. sp.). The online version of this work is archived and available from the following digital repositories: PeerJ, PubMed Central and CLOCKSS.

## RESULTS

A total of 8 *Caesio cuning* were examined, total length (TL) 15–27 cm. Of five fish from Pulau Ketam, one fish (TL 18 cm) was positive for a gastric myxosporean. From three fish in Langkawi, none were found to be infected. Therefore, only one fish from eight was infected by this myxosporean.

Myxospores had curved outlines in both sutural and valvular view; mean length 9.0 μm (range: 8.5–9.3 μm), width 5.7 μm (5.0–6.3 μm), thickness 6.0 μm (5.7–6.3) (Figs. 1 and 2; Table 1). In valvular view they are lemon-shaped to ovoid with tips on each end and symmetrical along both axes. The valves are identical, with nipple-like protrusions at the opposite end and side of the spore, giving it a sigmoid appearance and a rotational symmetry (180°) in the sutural view (Figs. 1 & 2). The spore valves lack striations (Fig. 3) but have a curved suture that bisects the spore near the polar capsules. Two large elongate polar capsules (PC), mean size 4.6 μm (4.3–4.9 μm) × 2.6 μm (2.2–3.1 μm), lie at opposite ends of the spore, somewhat overlapping each other in the valvular plane. Polar filaments have 7–9 coils and are discharged, in opposing directions, through the capsular foramina that are located at the pointed extremities of the spore valves. The sporoplasm is undivided and binucleate, filling most of the cavity surrounding the PC cells (Fig. 1).

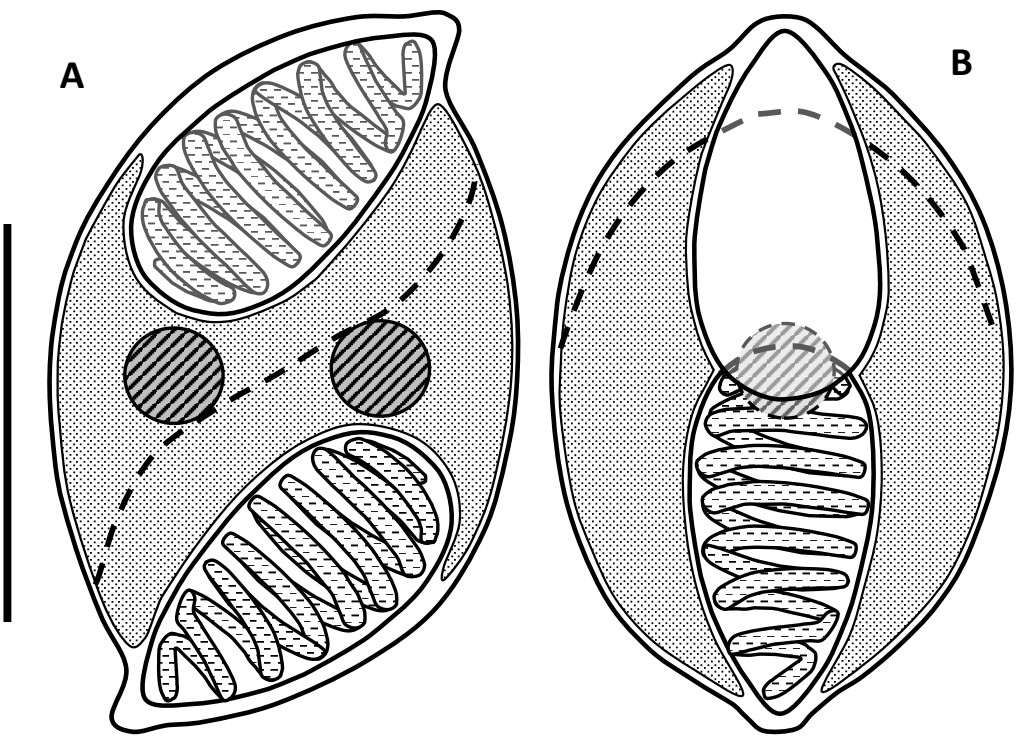

**Figure 1** **Line drawings of myxospores of *Enteromyxum caesio* n. sp.** (A) Spore shown in the sutural view. (B) Spore shown in valvular view. Position of the suture is shown with a dashed line, scale bar = 5 µm.

In histological section (Fig. 2A), large masses of disporous pseudoplasmodia were seen in what we assume to be the glandular part of the stomach wall, which had sloughed off post mortem, presumably as the fish were bought from a market and were fixed some hours after capture. Consequently, reliable description of host tissue response / pathology, associated with infections is not possible. SEM of myxospores confirm that the spore surfaces are smooth and not striated as in the Myxidiidae (s.s), with a non-conspicuous suture between the two valves (Fig. 3). Each valve has valvular projections or ridges adjacent to a terminal structure that houses the polar capsule (Fig. 3). TEM of a tiger puffer intestine infected with *E. fugu* shows developing epicellular stages of the parasite attached to the villi of the intestinal lining (Fig. 4).

A single SSU rDNA consensus sequence of 1,709 bp was successfully generated (GenBank accession number MT311171), BLAST searches revealed the highest identity matches were for the three currently described species of *Enteromyxum*, but only with an 83–84% similarity (Table 2). Phylogenetic analyses of myxosporeans representing the main marine clades, robustly grouped *E. caesio* with the other three species of *Enteromyxum*, irrespective of the phylogenetic methodology used (Fig. 5).

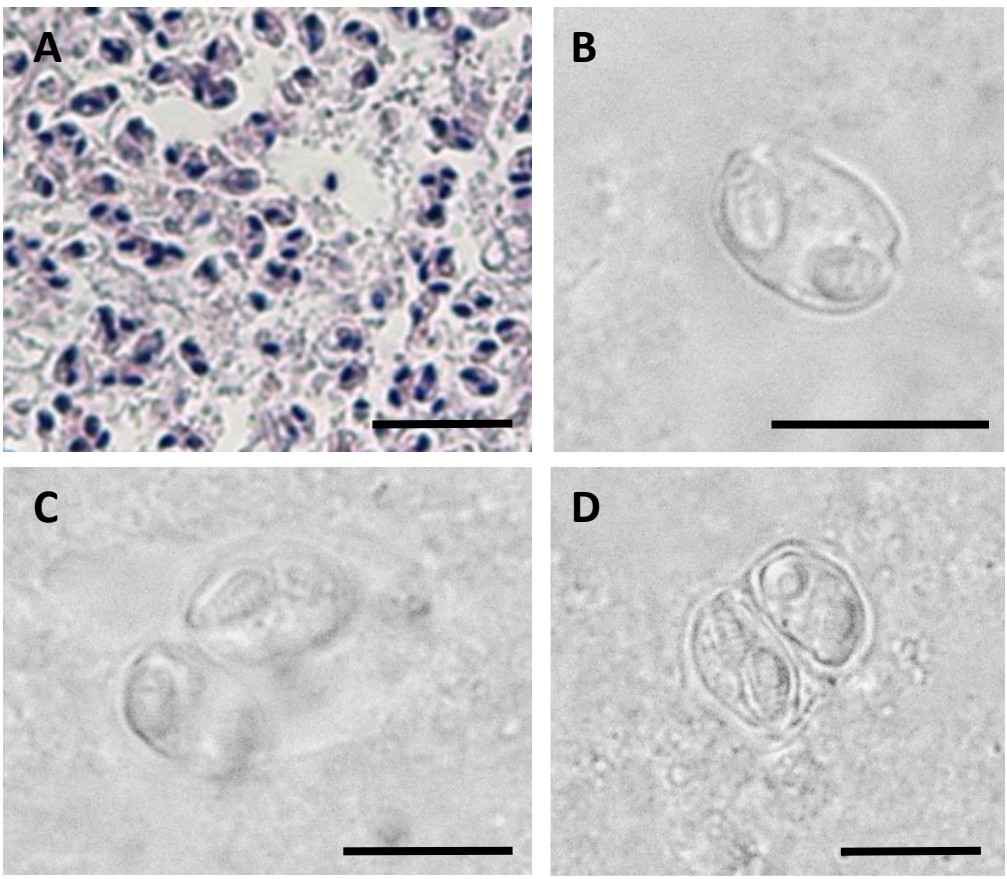

**Figure 2** **Myxospores of *Enteromyxum caesio* n. sp.** (A) Giemsa stained histological section from the stomach of *Caesio cuning*, detailing a large mass of disporous pseudoplasmodia of *E. caesio* n. sp. (B) Fresh spore in sutural view. (C & D) Disporous pseudoplasmodia showing the variation in spore morphologies seen dependent on the plane of focus. Scale bars A = 30 μm, B–D = 10 μm.

Class Myxozoa Grassé, 1970
Subclass Myxosporea Bütschli, 1881
Order Bivalvulida Shulman, 1959
Suborder Variisporina Lom et Noble, 1984
Enteromyxidae n. fam. Freeman, Yanagida et Kristmundsson, 2020

## Taxonomic summary

Maintaining characteristics of the generic description of *Palenzuela, Redondo & Alvarez-Pellitero (2002)*, updated to include epicellular forms and infection of the glandular tissues of the stomach.

  Histozoic, sometimes epicellular or with epicellular stages, parasites of the intestinal epithelium or gastric glandular tissues of the digestive system of marine fishes. Causes acute enteritis, cachexia and death in susceptible fish, entirely epicellular species are not pathogenic. Myxospores with slightly crescent shape, relatively large and elongated polar

Freeman et al. (2020), *PeerJ*, DOI 10.7717/peerj.9529

**Table 1  Myxospore dimensions reported for *Enteromyxum* spp.**

| Myxozoan species | Fish host/s | Spore dimensions (μm) | | | Polar capsules (μm) | | | Reference |
|---|---|---|---|---|---|---|---|---|
| | | Mean length (Range) | Mean width (Ran | Mean thickness | Mean length (Ra) | Mean width | Number of (c) | |
| | *Sparus aurata* | 14.7 (13.2–15.2) | 6.9 (5.6–7.8) | 6 | 7.4 (6.2–8.8) | 3.2 (2.8–3.8) | 7 (6–8) | *Diamant, Lom & Dyková (1994)* |
| *Enteromyxum leei* | *Diplodus puntazzo Takifugu rubripes* 14 | (15–19) | (5–7) | n.d. | 6.5-9.0 | 2.5–4.0 | n.d. | *Le Breton & Marques (1995)* |
| | *Sciaenops ocellatus* | 17.5 (15.5–19.5) | 7.4 (7.0–8.7) | n.d. | 8.4 (7.0–9.8) | 3.8 (3.3–4.5) | n.d. | *Diamant (1998)* |
| *Enteromyxum fugu* | *Takifugu rubripes* | 14.4 (13.5–15.5) | 9.0 (8.0–10.0) | n.d. | 6.1 (5.0–7.5) | 2.2 (2.0–3.0) | n.d. | *Tun, Yokoyama & Ogawa K Wakabayashi (2000)* |
| *Enteromyxum scophthalmi* | *Scophthalmus maximus* | 22.2 (20–25) | 11.7 (9.2–14.1) | 14 | 10.4 (8.6–13.0) | 4.6 (3.6–6.0) | 11 | *Palenzuela, Redondo & Alvarez-Pellitero (2002)* |
| *Enteromyxum caesio* n. sp. | *Caesio cuning* | 9.0 (8.5–9.3) | 5.7 (5.0–6.3) | 6.0 (5.7–6.3) | 4.6 (4.3–4.9) | 2.6 (4.1–4.9) | 8 (7–9) | Present study |

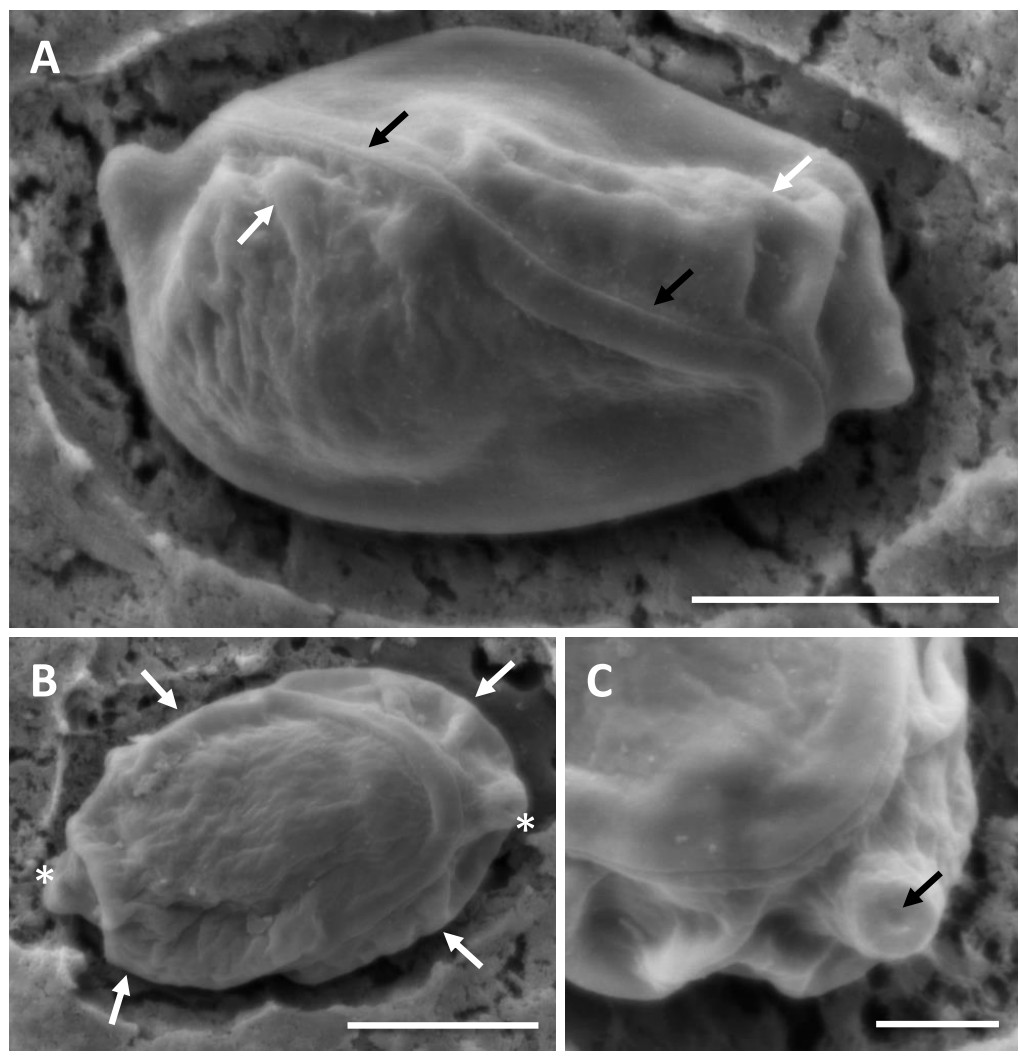

**Figure 3** **SEM of myxospores of *Enteromyxum caesio* n. sp.** (A) Sutural view; note the smooth non striated valves, flush suture (black arrows) and valvular ridges (white arrows). (B) Valvular view; note the ridges (white arrows) at edge of each valve, adjacent to the polar capsules (white asterisks). (C) High magnification of the tip of the polar capsule detailing the capsular foramina. Scale bars A & B = 3 μm, C = 1 μm.

**Table 2** **Percentage identities of SSU rDNA sequences from the *Enteromyxum* clade (Fig. 5) above 4 diagonal, and number of bases compared, below diagonal.**

| | | | | | |
|---|---|---|---|---|---|
| *Enteromyxum caesio* n. sp. | – | 82.29 | 82.15 | 83.66 | 79.07 |
| *Enteromyxum scophthalmi* | 1,572 | – | 86.40 | 88.65 | 84.58 |
| *Enteromyxum leei* | 1,569 | 1,573 | – | 84.31 | 83.03 |
| *Enteromyxum fugu* | 1,499 | 1,506 | 1,491 | – | 85.01 |
| Unicapsulactinomyxon | 994 | 999 | 990 | 994 | – |

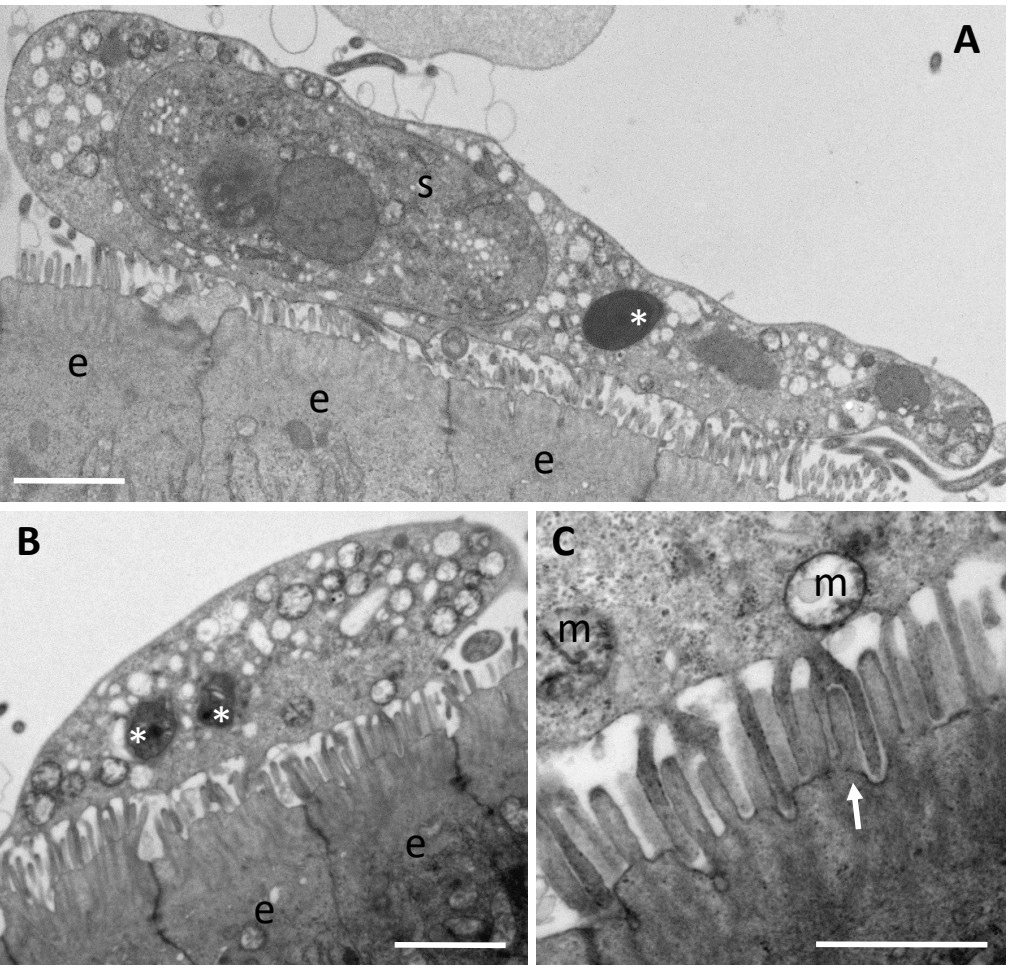

**Figure 4** **Trophozoites of *Enteromyxum fugu*, attached to the microvilli of the intestinal lining by pseudopodia.** (A) Large trophozoite containing a secondary cell, hyperinfected with a microsporidian spore (white asterisk). (B) Immature trophozoite containing many mitochondria and developing microsporidians (white asterisks). (C) Close up of the pseudopodia from (B) detailing their attachment around the entire microvillus (white arrow). (e) enterocyte, (m) mitochondria, (s) secondary cell. Scale bars A & B = 2 µm, C = 1 µm.

capsules tapering to their distal side and opening at the ends of the spore but discharging in opposite directions relative to a longitudinal plane bisecting the spore in top or bottom view. One binucleated sporoplasm. Spores develop in disporic pansporoblasts.

Single genus: *Enteromyxum* (*Palenzuela, Redondo & Alvarez-Pellitero, 2002*)

Type species. *Enteromyxum scophthalmi* (*Palenzuela, Redondo & Alvarez-Pellitero, 2002*)

Type host. *Scophthalmus maximus* (Linnaeus, 1758)

*Enteromyxum caesio* n. sp. Freeman, Yanagida et Kristmundsson, 2020

Type locality: Pulau Ketam, Peninsular Malaysia [3°5′35.98″N; 101°14′34.17″E]

Type host: *Caesio cuning* (Bloch, 1791)

Site of infection: histozoic in the stomach wall, likely in the glandular tissues
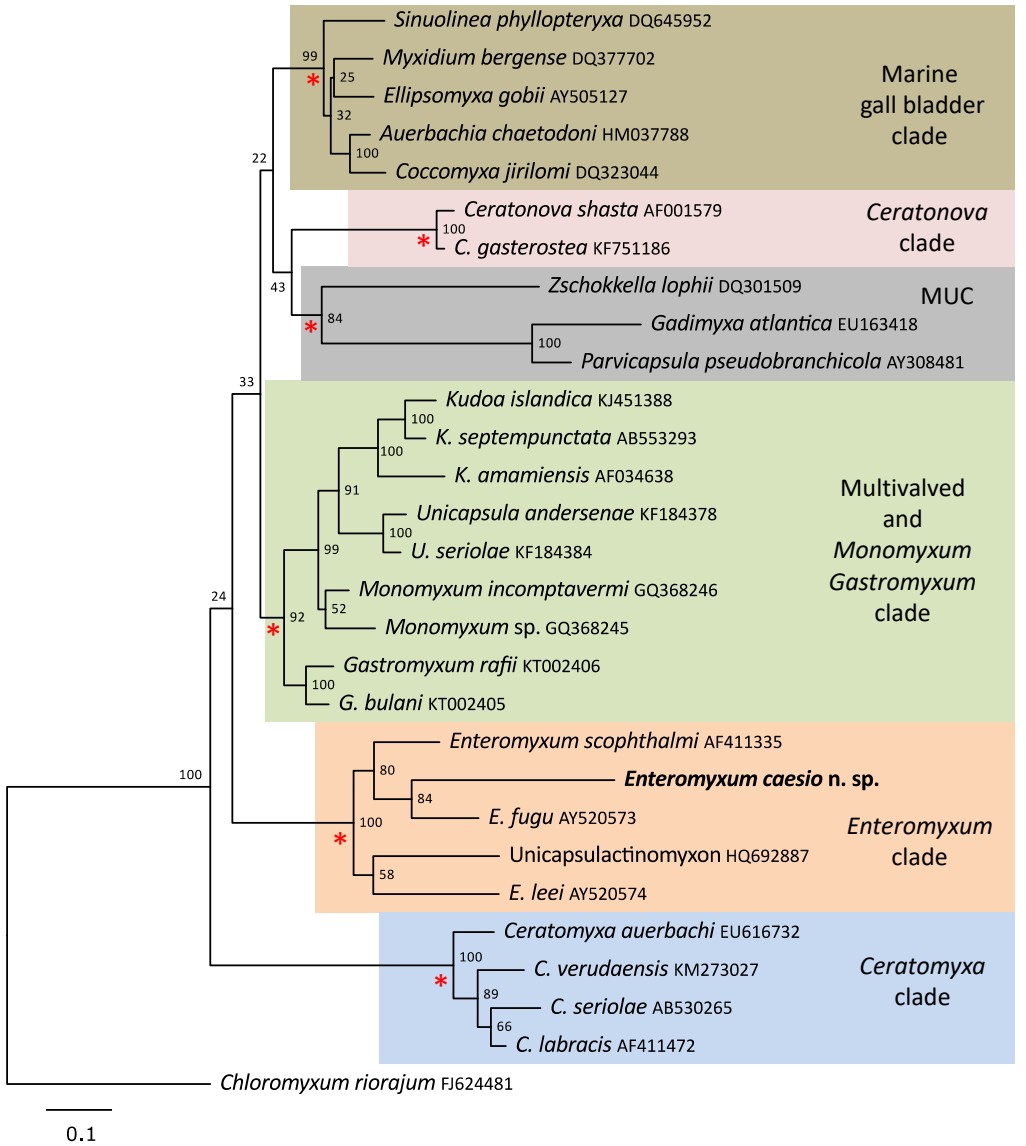

**Figure 5 Maximum likelihood topology of the main clades of marine myxosporeans.** The main marine myxosporean clades, shown in coloured boxes, are well supported from the relevant nodes (red asterisks). However, the relative grouping of these robust clades is not supported, apart from the most basal *Ceratomyxa* clade. *Enteromyxum caesio* n. sp. is robustly grouped within the *Enteromyxum* clade. *Chloromyxum riorajum* was used as an outgroup. Accession numbers are given after taxon names and numbers at the nodes represent bootstrap support from 1,000 resamplings. (MUC) Marine Urinary Clade.

Etymology: caesio refers to the scientific name of the host.

Type material: a 1709 bp SSU rDNA sequence with the accession number MT311171 has been submitted to NCBI.

## DISCUSSION

It has been previously demonstrated with independent molecular phylogenetic analyses that *Enteromyxum* spp. form a monophyletic clade within the marine myxosporeans that is distinct from members of the genus *Myxidium* (*Palenzuela, Redondo & Alvarez-Pellitero, 2002*; *Yanagida et al., 2004*; *Fiala et al., 2015*). Furthermore, it has been demonstrated that the histozoic genera *Enteromyxum*, *Gastromyxum* and *Monomyxum* form separate but robustly supported clades that allowed the creation of two new families Gastromyxidae (*Freeman & Kristmundsson, 2015*), and Monomyxidae (*Freeman & Kristmundsson, 2015*), to accommodate *Gastromyxum* spp. and *Monomyxum* spp. respectively (*Freeman & Kristmundsson, 2015*). It was also stated, by *Freeman & Kristmundsson (2015)*, that the placement of *Enteromyxum* in the family Myxidiidae was not considered correct and that a new family was warranted for the genus. As such, we have created a new family, Enteromyxidae n. fam., to accommodate this distinctive and important genus of myxosporean parasites and described the fourth species.

*Enteromyxum* share some similarities to *Gastromyxum* such as smooth valves with an indistinct suture bisecting the spore with a similar overall morphology, and both are histozoic in the digestive tracts of marine fishes. However, notable differences also exist, with respect to the size and extent on the polar capsules and the pathogenic nature of *Enteromyxum* compared to *Gastromyxum*, although *Gastromyxum* could potentially become pathogenic in an aquaculture setting. Perhaps the most significant and consistent difference between these genera is their reproducible placement within phylogenetic analyses of marine myxosporeans as robustly supported but separate clades (*Freeman & Kristmundsson, 2015*; *Casal et al., 2019*; present study). This remains the primary reason that this genus requires a distinct family as inclusion in the Gastromyxidae would create unacceptable paraphyly in what is a robust section of the phylogeny of histozoic marine myxosporeans. Although the relationship between the histozoic genera *Enteromyxum*, *Gastromyxum* and *Monomyxum* is consistently recovered as basal to the kudoids (*Freeman & Kristmundsson, 2015*; *Casal et al., 2019*), their relationship between other robust histozoic clades within the marine myxosporeans, such as the *Ceratonova* clade, are not always retrieved (*Fiala et al., 2015*). These inconsistencies make the higher-level grouping of such clades unreliable; until additional sequences are uncovered that allow such relationships to be more consistently demonstrated.

The overall myxospore morphology of *E. caesio* n. sp. is typical of those from the genus; however, the actual spore size is much smaller than those previously described (Table 1). Using myxospore morphology for diagnosis could result in confusion between other genera that are phylogenetically distinct but found in the digestive tract of marine fish, such as *Sigmomyxa, Gastromyxum* or some *Myxidium* spp., therefore, identification using molecular techniques is the preferred method to support initial diagnosis, especially considering the potential impact of *Enteromyxum* spp. to farmed marine fish.

Although myxospore morphology is well conserved within the *Enteromyxum* clade, the four described species are genetically divergent sharing only between 82.2%–88.7% identity to each other with respect to SSU rDNA sequences. This phenomenon has been

reported before, where a robustly supported clade of myxosporeans, the *Paramyxidium*, has a highly conserved myxospore morphology but a broad genetic diversity (*Freeman & Kristmundsson, 2018*). For the *Paramyxidium*, it has been suggested that this is due to a significant under-sampling of species within the genus/clade and that many more members of the *Paramyxidium* will be uncovered with more extensive molecular sampling (*Fiala et al., 2019*). This, along with the discovery of *E. caesio*, suggests that there could be many more species of *Enteromyxum* that are as yet undescribed, infecting wild fish that could impact the globally expanding marine finfish aquaculture industry.

There are reports that some species of *Enteromyxum* are epicellular (*Yanagida et al., 2004*; *Cuadrado et al., 2007*), and developing trophozoites are clearly seen attached to the intestinal villi via pseudopodia in *E. fugu* (Fig. 4). True epicellular species are not thought to cause the clinical signs of enteromyxosis, but any detached or sloughed off epicellular forms are likely to cause fish-to-fish transmission in intensive net pen aquaculture systems, albeit benign. In advanced histozoic infections of *E. leei*, fish develop enteritis and liquid filled intestines (*Sitjà-Bobadilla & Palenzuela, 2012*). During the excretion of such material, extensive sloughing of the intestinal wall is likely which leads to the release of developing histozoic stages into the environment. In intensive finfish culture, these stages are transmitted to cohabiting uninfected fish in the same cage and develop to cause the clinical signs of enteromyxosis (*Picard-Sánchez et al., 2020*). Mixed infections of histozoic species, like *E. leei*, with epicellular *E. fugu* and an unidentified epicellular myxosporean have been reported in aquaculture settings, but the histozoic species alone are considered to cause disease (*Yanagida et al., 2005*; *Cuadrado et al., 2007*).

Myxosporean stages were also observed in fresh smears or mucosal scrapings or in the fluid present in the digestive lumen of farmed turbot with clinical signs of disease (*Branson, Riaza & Alvarez-Pellitero, 1999*). *Branson, Riaza & Alvarez-Pellitero (1999)* also noted that the primary infection might be caused by a waterborne actinospore, as there was a random pattern of initial disease occurrence throughout affected farms. However, once infection was established the disease spreads rapidly, with 100% mortality in all tanks where it was first diagnosed, again suggesting rapid fish-to-fish transmission within the same tank. In our phylogenetic tree, there is a DNA sequence from actinospores released from a marine polychaete in Portugal (*Rangel et al., 2011*) that groups within the *Enteromyxum* clade and has typical inter species genetic variation for the genus (Table 2), suggesting that typical transmission to fish in this genus is via an actinospore.

## CONCLUSIONS

Herein, we have described a fourth species of *Enteromyxum*, *E. caesio* n. sp., and demonstrated the necessity for a new family, the Enteromyxidae n. fam., to accommodate this distinct and important genus of myxosporeans. Members of the Enteromyxidae n. fam. are parasites of the gastrointestinal tract of marine fishes and histozoic species can cause the disease enteromyxosis in intensive finfish aquaculture facilities. The Enteromyxidae n. fam. are phylogenetically related to other histozoic myxosporeans from the marine environment, but remain a discreet group. Within the Enteromyxidae n. fam., there is a

significant genetic range with respect to SSU rDNA sequence data, and it is expected that numerous other species in this family are yet to be discovered.

### Funding
This project was financially supported by a University of Malaya Research Grant RP001L-13SUS. The funders had no role in study design, data collection and analysis, decision to publish, or preparation of the manuscript.

### Grant Disclosures
The following grant information was disclosed by the authors:
University of Malaya Research Grant: RP001L-13SU.

### Competing Interests
The authors declare there are no competing interests.

### Author Contributions
- Mark A. Freeman conceived and designed the experiments, performed the experiments, analyzed the data, prepared figures and/or tables, authored or reviewed drafts of the paper, and approved the final draft.
- Tetsuya Yanagida and Àrni Kristmundsson performed the experiments, analyzed the data, prepared figures and/or tables, authored or reviewed drafts of the paper, and approved the final draft.

### DNA Deposition
The following information was supplied regarding the deposition of DNA sequences:
The sequence is available at GenBank: MT311171.

### Data Availability
The 18S DNA sequence is available in a Supplementary File.

### New Species Registration
The following information was supplied regarding the registration of a newly described species:
Publication LSID: urn:lsid:zoobank.org:pub:A8A08CD0-7323-4F4D-A8A9-E51FF86323CE
New Family (Enteromyxidae n. fam.) LSID: urn:lsid:zoobank.org:act:676E54CE-BF4C-4CEE-BEF1-83EA20E88510
New species (Enteromyxum caesio n. sp.) LSID: lsid:zoobank.org:act:2DE3D545-79D5-4D16-9932-2FC32E09E69E.

### Supplemental Information
Supplemental information for this article can be found online at http://dx.doi.org/10.7717/peerj.9529#supplemental-information.

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
