# Peer review of "A novel histozoic myxosporean, Enteromyxum caesio n. sp., infecting the redbelly yellowtail fusilier, Caesio cuning, with the creation of the Enteromyxidae n. fam., to formally accommodate this commercially important genus"

_PeerJ, doi:10.7717/peerj.9529_

## Round 0.1 · original submission · Major Revisions

Your manuscript has major concerns. The Latin name of the species mentioned have to be added in all the text. The histological figures should be improved. Other comments from the reviewers are provided.

Reviewer 1 ·

Basic reporting

Regarding the language, the manuscript is technically correct and clearly written and only a few minor mistakes have escaped the author’s corrections.
- L149-159: Units should be separated by one space from numbers.
- L168: Please, correct verb conjugation or tense.
- L272: Future tense should be replaced by present tense.
- Table 1: “Length Mean”, “Width Mean” and “Thickness Mean” should be replaced by “Mean Length”, “Mean Width” and “Mean Thickness”.
- Table 1: Spelling of “Scophthalmus maximus” should be corrected.

There are further minor errors regarding the content of the manuscript, which should also be reviewed.
- Table 1: Polar filament coils are not given in µm.
- Table 1: Means are missing for myxospore dimensions of E. leei in D. puntazzo.
- Figure 2: The authors claim to have found a new parasite species in the “glandular gastric tissue of the stomach”, but no host tissue or cells are recognizable in the image shown. A lower magnification image would help to locate the parasite stages in the organ.
- Figure 2: Does the insert belong to the Giemsa-stained histological section? It should.
- Figure 3: There is no “black scale bar”. Please correct.
- Figures: I would suggest maintaining a consistent scale bar format, i.e. indicating scale bar length either in the figure caption or in the image.
- References: Please, check reference order (e.g. Diamant A 1997; Diamant A, Lom J, Dyková I 1994; Diamant A 1998). Please, references of same authors and years should be identified by letters as different citations in the text and in the reference list (e.g. Freeman MA, Krismundsson Á 2015).

Experimental design

The first issue regarding the present experimental design concerns the method used to identify infected fish: field observation of wet mounts of stomach and intestinal scrapings. This method is a fast and easy one to determine the presence of Enteromyxum spp. only in advanced, high intensity infections. Pre-sporogonic stages are difficult to be identified in this way, and even more at initial infection stages when parasite stages are small or even unicellular. Furthermore, authors do not give any explanation why stomach and intestine were chosen for this, while esophagus and rectum were ignored, nor which intestinal segment (anterior, middle, posterior, entire?) was used.
Therefore, the presence of the parasite in the examined fish might have been underestimated, and thus, authors limited their sample size to just one infected fish.

There is a further issue, mainly, the authors are claiming to describe for the first time E. caesio n. sp. in redbelly yellowtail fusilier and they included in the manuscript TEM images of another parasite of the same genus.
Though images (Figure 4) show nicely the host-parasite interaction at cellular level in E. fugu-infected tiger puffer, as was previously demonstrated in E. leei-infected sea bream (Cuadrado et al. 2007), this is not relevant to the research work presented herein. The manuscript’s title also excludes the significance of this result by itself and its relevance for the E. caesio n. sp. description by omitting to mention it and it is a surprise for the reader when it shows up in the manuscript since it has no relationship with the main matter. My suggestion is to eliminate the results on E. fugu from the manuscript, since the aim of this work as it is now, is not the characterization of host-parasite interactions. These data would be very interesting for further research on fish-myxozoan interactions.

Materials & Methods
This section is lacking important details about the animals used and their sampling. Some of the missing information (e.g. fish numbers and size) is surprisingly provided later in the results section.
- L83: “Fish were bought… others were captured” is too ambiguous. Fish species, amount, size and if dead or alive should be indicated here. If captured fish were collected alive, authors should describe how the sacrifice was performed.
- In this regard, this reviewer ignores the legal issues of lethally sampling wild animals for scientific purposes in Saint Kitts and Nevis, but the authors do not mention in their manuscript any bioethics committee approval or adaptation of their methods in accordance to the local legislation. An ethics statement containing such information has to be provided in this section.
- L87: Authors declare to have taken images of fresh myxospores, but they do not show them. In addition to my previous comments on Figure 2, I would suggest to include a wet mount image of parasite stages (not only as an insert).
- L88-89: “Infected tissue samples…” Please, define which tissues. I would strongly suggest examining the entire digestive tract (from esophagus to rectum).
- L92: Again, authors should mention fish origin, number, size and method used for their sacrifice.
- L96: Authors should clarify in the text, which fish they used to obtain these intestinal scrapings.
- L149: “ A total of 8 fish…” Authors should indicate which fish species they are talking about since they worked with fusiliers and puffers.

Validity of the findings

The main drawback of the present work is that the entire research is based on the one and only infected fish sampled, due to the low infection prevalence found and to the small sample size.
I wonder how convenient and valid the taxonomic redistribution and the new species description are, which the authors suggest based on the SSU rDNA sequence they obtained from one single sample.
However, molecular phylogeny is not my area of expertise, and thus, I leave this matter up to other reviewers’ opinion or editorial decision.

There is a further important issue that needs to be improved by the authors. Histological samples should be taken immediately after death, but in the case of the fusilier’s samples tissues were fixed “some hours after being caught”. Though authors state that therefore, “reliable description of host tissue response was not possible”, they affirm, “disporous pseudoplasmodia were seen in the glandular part of the stomach wall (gastric glands)”. The main issue is that, at least from the present image (Figure 2), the cellular host response as well as the location of the parasite stages at tissue level is impossible, as I mentioned before. Only by providing histological images in which the tissue structure is discernible, the parasite presence in the gastric glands would be confirmed. Otherwise, histological samples have the same validity as fixed and stained tissue scrapings or squashes. Therefore, the obtaining of life, infected fish is crucial if authors aim to describe tissue location of the parasite. A further suggestion once this is achieved, is to include ultrastructural observations by TEM in their study.

Additional comments

The present results aim to define a new species within the Enteromyxum genus and the erection of a new taxonomic family in order to accommodate members of this genus separated from Gastromyxidae and Myxidiidae. Nevertheless, the quality and amount of samples used therefore should be improved in order to validate this very interesting and promising data, which seem still preliminary.

Reviewer 2 ·

Basic reporting

This study reports a novel Enteromyxum sp. in the stomach of the redbelly yellowtail fusilier in Malaysia as Enteromyxum caesio n. sp., and erects a new family, Enteromyxidae n. fam., for the genus Enteromyxum, currently containg four species. The manuscript is well constructed and logically described with clear figures. The reviewer has no indications or suggestions except for a few minor points as mentioned below:

Experimental design

No problem.

Validity of the findings

No problems.

Additional comments

The reviewer has no indications or suggestions except for a few minor points as mentioned below:

1. L1 [Title]: ‘sp.’ after ‘Enteromyxum’ (?)
2. L19: Is it necessary put ‘,’ after ‘mortalities’?
3. L25, L30, L35, L77, L225: ‘Enteromyxidae n. fam.’
4. L26, Figure 2 legend: Either ‘gastric’ or ‘of the stomach’ might be enough.
5. L34: ‘spore valve’ or ‘shell valve’ (?)
6. L35: ‘is’ (?)
7. L44: Full description of scientific names at their first appearance might be welcome by the readers: ‘the genus Enteromyxum Palenzuela, Redondo et Alvarez-Pellitero, 2002’.
8. L46, L50: The scientific name for the turbot and tiger puffer.
9. L61, L235, L240: Two references of ‘Freeman & Kristmundsson, 2015’ are found in the Reference section.
10. L68-69: Please place references chronologically.
11. L71: ‘Caesio cuning (Bloch, 1791)’ (?)
12. L83: What are ‘othres’?
13. L87, L88, L95: ‘Dino-Eye’, ‘ImageJ’, and ‘Jeol JEM-1010’ are not familiar to the readers.
14. L93: ‘PCR positive’ is too much simple expression for the readers.
15. L96: ‘DNA lysis buffer’ (?).
16. L83-L128: The reviewer recommend the authors to follow the basic manner of explaining some commercially available products as usual in the scientific articles.
17. L184-L186: ‘Order Bivalvulida Shulman, 1959’ (?) What is the reason for you ro omit it?
18. L186-L188, L200-K202: If you use ‘et’ here, it is better to use ‘et’ for scientific names at any times. If you can use ‘& (and)’ for this purpose, use it thoroughly in this manuscript.
19. L191: Does Enteromyxum caesio n. sp. prefer to parasitize only cardiac gastric glandular tissues only? How about other places of the stomach? Just for confirmation.
20. L203: Full scientific name.
21. L221: Full expression of Gastromydicae and Monomyxidae.
22. L257: ‘Freeman & Kristmundsson, 2018’ not in the Reference section.
23. L353: Finally, the reviewer understands that ‘2018’ must be placed here instead of ‘2015’. Please check citations of ‘Freeman & Kristmundsson, 2015’ and ‘Freeman & Kristmundsson, 2018’ thoroughly in the text.
24. L389-390: ‘Scomphthalmus maximus’ in italic (?)
25. L313: ‘Enteromyxum species.’
26. Table 1: ‘Tin Tun et al. (2000)’ (?)
27. Figure 4: Is it possible for the reviewer to see a high magnification TEM photographs of hyperinfected microsporidia just for review purpose? At low magnification, it seems a homogenous structure.
28. Please check thoroughly the manuscript by themselves.

This manuscript is very interesting and valuable for us.

Annotated reviews are not available for download in order to protect the identity of reviewers who chose to remain anonymous.

Reviewer 3 ·

Basic reporting

The article meets all the criteria.

Experimental design

The article meets all the criteria.

Validity of the findings

The article meets all the criteria.

Additional comments

The present paper describes a single myxosporean species assigned to the genus Enteromyxum. Authors provide morphological documentation of the parasite including also SEM and TEM ultrastructure details of the spores and trophozoites, respectively, as well as phylogenetic analysis of the studied myxosporean species and its relatives.
The main message of the manuscript is a species description of another species of important genus Enteromyxum, which member E. leei is a serious pathogen of cultured fish. In addition, authors erected a new family, Enteromyxidiidae, to accommodate four members of the genus Enteromyxum. Taxonomic changes within Myxozoa are challenging and have to be done very cautiously because of the presence of many paraphyletic and polyphyletic systematic groups. I agree with a new family Enteromyxidiidae - its establishment is another step to a better myxozoan taxonomy.
Specific comments:
The title of the manuscript should contain scientific Latin name of the fish host (English common names are not sufficient). I would also suggest adding higher classification for Enteromyxum (Myxozoa) in the title (not all parasitologists are aware of what group of parasite Enteromyxum is).
Similarly, in M&M, there should be a Latin name for fish host species provided. Please, provide also GPS coordinates for Kilim mangoves, Langkawi (is there a typo? “mangoves“ = “mangroves”).
Taxonomic summary, line 180-187: According to a book Myxozoan Evolution, Ecology and Development (Okamura et al. 2015), the taxonomic classification of Myxozoa is unranked subphylum, Myxosporea should be a class. There is an order Bivalvulida missing in the summary.
The histology section, as well as spore light microscopy, is of very poor quality in Fig.2. Do you have any better images?

---

## Round 0.2 · accepted · Accept

I am pleased to confirm that your paper has been accepted for publication in PeerJ.

Thank you for submitting your work to this journal.

Reviewer 1 ·

Basic reporting

The article meets all the criteria.

Experimental design

The article meets all the criteria.

Validity of the findings

The article meets all the criteria.

Additional comments

The authors have improved the manuscript significantly, as well as clarified all issues.

Reviewer 3 ·

Basic reporting

no comment

Experimental design

no comment

Validity of the findings

no comment

Additional comments

I am satisfied with the author's responses and changes made in the manuscript.